# Effect of Aflatoxin B1 on the Nervous System: A Systematic Review and Network Analysis Highlighting Alzheimer’s Disease

**DOI:** 10.3390/biology14040436

**Published:** 2025-04-17

**Authors:** Samira Ranjbar, Pantea Mohammadi, Somayeh Pashaei, Masoud Sadeghi, Masomeh Mehrabi, Sasan Shabani, Ali Ebrahimi, Annette B. Brühl, Reza Khodarahmi, Serge Brand

**Affiliations:** 1Medical Biology Research Center, Health Technology Institute, Kermanshah University of Medical Sciences, Kermanshah 6714415185, Iran; samira.ranjbar64@yahoo.com (S.R.); panteamohammadi18@yahoo.com (P.M.); bahar.1950@gmail.com (S.P.); sadeghi_mbrc@yahoo.com (M.S.); mehrabimasomeh@yahoo.com (M.M.); sasan.shabani96@gmail.com (S.S.); 2Department of Clinical Biochemistry, School of Medicine, Kermanshah University of Medical Sciences, Kermanshah 6714415185, Iran; 3Dermatology Department, Hajdaie Dermatology Clinic, School of Medicine, Kermanshah University of Medical Sciences, Kermanshah 6714415185, Iran; 4Center for Affective, Stress and Sleep Disorders, Psychiatric Clinics, University of Basel, 4002 Basel, Switzerland; annette.bruehl@upk.ch; 5Department of Pharmacognosy and Biotechnology, Faculty of Pharmacy, Kermanshah University of Medical Sciences, Kermanshah 6714415185, Iran; 6Sleep Disorders Research Center, Kermanshah University of Medical Sciences, Kermanshah 6715847141, Iran; 7Substance Abuse Prevention Research Center, Kermanshah University of Medical Sciences, Kermanshah 6715847141, Iran; 8Division of Sport Science and Psychosocial Health, Department of Sport, Exercise and Health, University of Basel, 4002 Basel, Switzerland; 9School of Medicine, Tehran University of Medical Sciences, Tehran 1416753955, Iran; 10Center for Disaster Psychiatry and Disaster Psychology, Center of Competence for Disaster Medicine, Swiss Armed Forces, 4002 Basel, Switzerland

**Keywords:** AFB1, Alzheimer’s disease, angiopathy, inflammation, neurotoxicity, oxidative stress

## Abstract

Alzheimer’s disease is a serious condition that causes memory loss and problems with thinking and behavior. One harmful substance called aflatoxin B1, commonly found in contaminated food like nuts and grains, may play a role in making this disease worse. In this study, we reviewed existing research to understand how aflatoxin B1 affects the brain. We found that aflatoxin B1 damages DNA and creates stress inside cells, which leads to inflammation and harm to brain cells. It also disrupts important cell functions and may help form harmful protein clumps seen in people with Alzheimer’s disease. Our analysis also identified hundreds of genes related to this process, including key ones that could be targets for future treatments. This work shows how exposure to environmental toxins like aflatoxin B1 could increase the risk of Alzheimer’s disease. Understanding this connection can help scientists find new ways to protect the brain, detect problems earlier, and develop better treatments to slow down or prevent the disease.

## 1. Introduction

Neurodegenerative disorders are mainly characterized by the loss of neurons. The most common of these disorders include Alzheimer’s and Parkinson’s diseases [1]. Alzheimer’s disease (AD) is a chronic neurodegenerative disorder that significantly affects memory, spatial orientation, and cognitive functions [2,3,4,5,6,7]. It is recognized as the leading cause of dementia among older adults, particularly those aged 65 and over, and has a substantial global prevalence [7]. In Europe, the prevalence of AD is estimated at 5.05%, with a prevalence of 3.31% in males and 7.13% in females, increasing with age [8].

First-degree relatives of individuals with AD have a higher lifetime risk of developing the condition compared to the general population [9]. Genetic factors account for approximately 60–80% of the risk for AD, with over 40 associated genetic risk loci identified. Emerging biomarkers, such as PET scans and plasma assays for amyloid-beta (Aβ) and phosphorylated tau, hold significant potential for both clinical applications and research [10]. Several approved medications can alleviate certain symptoms of AD, but no existing treatments can alter its underlying disease mechanisms [11].

The fungi *Aspergillus* spp., which produce aflatoxins (AFs), are widespread in nature and have significantly contaminated human and animal food supplies, leading to health hazards and even fatalities [12]. There are over 20 known types of AFs, with the most well-known being B1, B2, G1, G2, M1, M2, aflatoxicol, and AFQ1 [13]. AFM1 and AFM2 are metabolites of AFB1 and AFB2, found in the milk of lactating mammals that consume AF-contaminated feed [13,14,15]. AFB1 is also strongly associated with various toxicities, including growth impairment, malnutrition, and immunomodulation [16]. AFB1 exhibits a range of biological activities, such as acute toxicity, teratogenicity, mutagenicity, and carcinogenicity [17]. AFB1 stands out as the most toxic compound among the identified AFs [18].

Once toxins cross the blood–brain barrier (BBB), they can directly impact glial cells and change the activation state of microglia and astrocytes, leading to brain inflammation, BBB disruption, and affecting the synaptic transmission process [19]. The presence of AFB1 in the brain tissue of children who died from Kwashiorkor disease suggests that AFB1 has a high ability to penetrate the BBB [20]. AFB1 can cross the BBB and induce cytotoxic effects in the microvascular endothelial cells of the BBB, leading to brain tissue damage [19,20,21] and affect astrocytes, microglia, and neurons [21,22,23].

Despite the conflicting data and lack of consensus regarding the types of cytokines induced by AF exposure, the upregulation of the immune response stimulates the production of tissue-damaging inflammatory molecules and free radicals. This leads to chronic inflammation, cancer, and neurodegenerative diseases [22,24,25,26,27,28]. AF–protein adducts are most frequently associated with acute intoxication, as they block protein synthesis, particularly the enzymes involved in vital functions such as metabolic pathways, protein synthesis, DNA replication and repair, and immune response [29,30,31,32]. Severe DNA fragmentation upon exposure to high doses of aflatoxins is a major effect of acute aflatoxicosis [33]. Aging and genetic factors are significant contributors to the onset of AD. However, certain environmental factors, such as chronic exposure to toxins, have also been reported to increase the risk of developing AD [34].

While previous reviews [18,19] have explored the neurotoxic effects of aflatoxins, few have systematically linked AFB1 exposure to the development of AD. Additionally, there remains a lack of understanding regarding the gene–environment interactions that may increase the risk of AD in individuals exposed to AFB1 [35]. This review aims to fill this gap by providing a systematic analysis of the effects of AFB1 on the nervous system, with a particular focus on AD. Our approach considers the impact of AFB1 on neural pathways, providing a comprehensive understanding of how AFB1 may contribute to the molecular basis of AD.

## 2. Materials and Methods

### 2.1. Study Design

This systematic review was conducted in accordance with the Preferred Reporting Items for Systematic Reviews and Meta-Analyses (PRISMA) guidelines [36]. The PECO question was as follows: Can AFB1 affect the nervous system, especially AD, based on human and animal models? The systematic review was not registered in any database. The study was registered in the PROSPERO database (registration number: CRD420250651007). Figures of pathways were created using CorelDraw Graphic Suite 2018 (*Corel Corp.* (*Ottawa, ON, Canada*)).

### 2.2. Study Search and Selection

A comprehensive search was conducted in the databases of Scopus, Cochrane Library, PubMed, and Web of Science up to 1 June 2024 without any restrictions. Initially, the titles and abstracts were screened, followed by the download of full-text articles based on the inclusion and exclusion criteria. The search keywords used were (“nerve” or “nervous” or “neuro*” or “CNS”) and (“aflatoxin*”). Additionally, the reference lists of the retrieved reviews, as well as “Google Scholar” and “ScienceDirect,” were reviewed to ensure no relevant study was missed.

### 2.3. Eligibility Criteria

We included any study reporting the effect of AFB1 in the nervous system, with the exception of book meta-analyses, chapters, reviews, conference papers, and bioinformatic analyses.

### 2.4. A Compilation of Interconnected Genes Linked to AD

The terms “Alzheimer’s disease”, “Inflammation”, “Angiopathy”, and “AFB1” were entered into the GeneCards database, which returned 1140 human genes linked to them.

### 2.5. Evaluation of Essential Target Genes, Along with Ontology (GO) and Kyoto Encyclopedia of Genes and Genomes (KEGG) Analysis

We utilized Excel software 2010 to identify the common genes associated with the terms mentioned above by employing its “sets to highlight duplicate items” feature. Subsequently, we used Cytoscape software 3.6.1 to develop the physical interaction network. Key targets were evaluated based on the metrics of “Degree”, “Closeness”, and “MCC”. GO and KEGG enrichment analyses were conducted to ascertain the molecular functions and systemic roles of the target genes.

## 3. Results

### 3.1. Study Selection Process

Figure 1 outlines the process of selecting studies for a systematic review or meta-analysis. It starts with identifying records through the databases (993 records), resulting in 718 records after removing duplicates. These records were then screened, leading to 33 full-text articles assessed for eligibility after removing 685 irrelevant records (not reporting the effect of AFB1 in the nervous system). Then, 17 records were excluded for various reasons (bioinformatics analyses, treatment studies, reviews, and studies without focusing on AB1). At last, 16 articles [23,37,38,39,40,41,42,43,44,45,46,47,48,49,50,51] were entered into the systematic review.

### 3.2. Characteristics of the Articles

Recent studies have investigated the effects of AFB1 on brain cells across different countries (study locations), revealing significant neurotoxic effects on various cell types (such as neurons, astrocytes, and microglia) in non-primate animal models and human cell lines, combining these ideas in vivo and in vitro (Table 1). The main results indicate that AFB1 exposure leads to DNA damage, oxidative stress, and apoptosis, with studies reporting increased lipid peroxidation, inflammation (elevated IL-6, NF-κB, TNF-α), and neurotoxicity, including α-synuclein pathology and dopaminergic neuron damage. Additionally, AFB1 disrupts metabolic enzymes like Na^+^/K^+^-ATPase and cyclic nucleotide phosphodiesterase, interferes with cell cycle regulation via p53 and p21, and induces mitochondrial dysfunction. Findings across species (human, rodent, fish, and livestock) highlight AFB1’s broad neurotoxic impact, reinforcing its potential role in neurodegenerative diseases and the need for further investigation into its long-term neurological effects.

### 3.3. Pathways

AFB1, radiation sensitive protein 51 (RAD-51) Oligomer, *poly [ADP-ribose] polymerase 1* (PARP-1), and breast cancer gene 2 (BRCA-2) components are crucial in maintaining genomic integrity by repairing DNA damage (Figure 2). In the context of AD, impaired DNA repair mechanisms can lead to neuronal damage and cognitive decline. AFB1 causes DNA adducts, which are harmful modifications to the DNA structure [47,48]. BRCA-2 binds to RAD-51 Oligomer and facilitates its attachment to the damaged DNA site. This complex then initiates the DNA repair process, helping to correct the damage caused by AFB1. Essentially, BRCA-2 plays a crucial role in recruiting RAD-51 to the site of damage, enabling the repair of AFB1-induced DNA lesions [42]. PARP-1 catalyzes the addition of poly (ADP-ribose) chains to itself and other proteins, signaling for the assembly of DNA repair components [42]. This process is crucial for repairing the DNA damage caused by AFB1. However, if the damage is extensive or the PARP-1 function is compromised, the repair process may be insufficient, leading to genomic instability and an increased risk of cancer development [37,42]. This highlights the importance of PARP-1 in maintaining genomic integrity and its potential as a therapeutic target in conditions involving DNA damage.

AFB1 induces significant endoplasmic reticulum (ER) stress, a condition marked by the accumulation of misfolded or unfolded proteins in the ER lumen. In response, the cell activates the unfolded protein response (UPR) to restore ER homeostasis. AFB1 specifically triggers two major arms of the UPR: the PERK (Protein kinase RNA-like ER kinase) and IRE1 (Inositol-requiring enzyme 1) pathways [50] (Figure 3).

Upon activation, PERK phosphorylates eIF2α, leading to a global reduction in protein synthesis to relieve the folding load. However, this also selectively promotes the translation of ATF4 (Activating Transcription Factor 4), which translocates to the nucleus and upregulates genes involved in oxidative stress, autophagy, amino acid metabolism, and apoptosis. Sustained ATF4 expression contributes to the upregulation of CHOP (C/EBP Homologous Protein), a pro-apoptotic transcription factor that inhibits anti-apoptotic Bcl-2 and enhances the expression of pro-apoptotic genes, thereby promoting programmed cell death [50].

Simultaneously, IRE1 undergoes autophosphorylation and splices XBP1 (X-box binding protein 1) mRNA, producing a potent transcription factor (XBP1s or X-box binding protein 1) that upregulates genes related to protein folding, ER-associated degradation (ERAD), and lipid biosynthesis. However, prolonged IRE1 activation also recruits TRAF2 (TNF receptor-associated factor 2), which leads to the activation of JNK (c-Jun N-terminal kinase) signaling [48].

Activated JNK phosphorylates BAX, a pro-apoptotic member of the Bcl-2 family, promoting its translocation to the mitochondrial outer membrane. This triggers the release of other pro-apoptotic proteins, such as BAK and BIM, culminating in mitochondrial outer membrane permeabilization (MOMP), cytochrome c release, and caspase cascade activation, leading to intrinsic pathway-mediated apoptosis [42,45,46]. Together, the PERK–ATF4–CHOP and IRE1–JNK–BAX signaling axes converge on mitochondrial dysfunction and apoptosis. The chronic activation of these pathways due to AFB1 exposure not only impairs neuronal viability but also disrupts protein homeostasis and energy metabolism [50]. These pathological changes accelerate neuronal loss, synaptic dysfunction, and neuroinflammation, all of which are key contributors to neurodegeneration and the progression of AD [41]. Furthermore, ER stress-induced apoptosis may amplify the accumulation of amyloid-beta (Aβ) and tau hyperphosphorylation, reinforcing the link between AFB1-induced ER stress and hallmark AD pathology [23,45,46].

AFB1 increases the expression and activity of Protein Kinase C (PKC), a key signaling molecule involved in various neuronal processes. Elevated PKC levels enhance Tyrosine Hydroxylase (TH) activity—the rate-limiting enzyme in dopamine biosynthesis—leading to an overproduction of dopamine (Figure 4). While dopamine is essential for normal neuronal function, its excessive accumulation can result in oxidative stress, neurotoxicity, and mitochondrial dysfunction, all of which contribute to neurodegeneration and the progression of disorders such as Parkinson’s disease and AD [41,46]. Moreover, dopamine oxidation generates reactive oxygen species (ROS), which further damages neuronal structures and exacerbates inflammatory responses [39,42].

Serotonin (5-HT) also upregulates PKC, reinforcing TH activity and dopamine synthesis [51]. Beyond its dopaminergic interaction, serotonin is implicated in the UMAD (Ubiquitin-Mediated Apoptotic Degradation) pathway, which plays a role in protein homeostasis and the degradation of misfolded proteins. Dysregulation of this pathway may lead to the accumulation of toxic protein aggregates in neurons.

A critical downstream effect of these imbalances is the misfolding and aggregation of alpha-synuclein (α-syn), a presynaptic neuronal protein. PKC overactivation, oxidative stress, and dopamine dysregulation collectively promote α-syn misfolding. Aggregated α-syn forms toxic oligomers and fibrils, which interfere with synaptic vesicle trafficking, impair neuronal communication, and trigger mitochondrial dysfunction [41]. These aggregates have been directly linked to TH-positive neuron degeneration and are hallmark features in diseases like Parkinson’s and AD.

Furthermore, α-syn aggregates are known to interact with amyloid precursor protein (APP) processing, facilitating the formation of amyloid-beta (Aβ) plaques and promoting tau hyperphosphorylation, which leads to neurofibrillary tangles—two pathological hallmarks of AD [42,46]. Thus, the combined effects of increased PKC signaling, dopamine dysregulation, serotonin involvement, and α-synuclein aggregation form a complex network that accelerates neuroinflammation, proteinopathy, and neurodegeneration, ultimately contributing to the onset and progression of AD.

AFB1 significantly impacts the brain by promoting oxidative stress and impairing the antioxidant defense system, as illustrated in Figure 5. Exposure to AFB1 leads to a marked increase in ROS, resulting in cellular oxidative damage [39,42]. This is accompanied by elevated levels of oxidative stress biomarkers, including malondialdehyde (MDA), lactate dehydrogenase (LDH), and lipid peroxidation (LPO) products [39]. Additionally, the accumulation of protein carbonyls and a reduction in thiol groups indicate extensive oxidative modification of proteins and lipids, which compromises neuronal function and integrity [46]. One of the major contributors to this ROS surge is the activation of NADPH oxidase 2 (NOX2), an enzyme complex that directly generates superoxide radicals [42]. Persistent NOX2 activation amplifies oxidative stress, creating a pro-inflammatory environment in the brain.

This oxidative burden is tightly linked with an inflammatory response, as ROS stimulate various intracellular signaling pathways that lead to the release of pro-inflammatory cytokines such as TNF-α, IL-1β, and IL-6 [39,46]. The resulting neuroinflammation disrupts the blood–brain barrier and facilitates the infiltration of immune cells into the central nervous system, exacerbating neural injury. Simultaneously, mitochondrial dysfunction becomes evident through significantly reduced ATP production, indicating a breakdown in cellular energy metabolism [41]. Ion homeostasis is also impaired, with altered levels of essential ions such as potassium (K⁺) and magnesium (Mg^2^⁺), further disrupting neuronal excitability and signal transmission [39]. These imbalances contribute to a cellular environment that is highly susceptible to apoptosis and neurodegeneration.

AFB1 also depletes the brain’s antioxidant defenses, weakening the activities of key enzymes such as superoxide dismutase (SOD), catalase, and glutathione peroxidase (GSH-Px) [39,46]. These enzymes play critical roles in neutralizing ROS and maintaining redox balance. When their activity is suppressed, oxidative damage escalates unchecked, leading to mitochondrial membrane damage, cytochrome c release, and activation of the intrinsic apoptotic pathway [42]. Ultimately, these molecular disruptions culminate in neuronal cell death, which impairs cognitive function and memory. Over time, such changes contribute to the development of AD, characterized by chronic neuroinflammation, synaptic loss, and the accumulation of pathological protein aggregates such as amyloid-beta plaques and neurofibrillary tangles [41,46]. AFB1, therefore, acts as a potent neurotoxin that initiates a cascade of oxidative and inflammatory events, driving the progression of neurodegenerative disorders.

### 3.4. Screening and Analyzing the Genes Shared by “Alzheimer’s Disease”, “Inflammation”, “Angiopathy”, and “AFB1”, as Well as Performing Network Analysis

The terms of “Alzheimer’s disease”, “Inflammation”, “Angiopathy”, and “AFB1” were entered into the GeneCards database, which returned 1140 human genes linked to them. From the 1140 related genes obtained, we screened out 309 genes shared with the mentioned terms. Then, we analyzed these 309 genes by Cytoscape and received a network diagram with multiple clusters (Figure 6).

To select the most important part of these 309 nodes, we screened them with centrality criteria like Degree, betweenness, and MCC. In a network graph, ESR1 had the most direct links with the other nodes. At the same time, TIMP3 had the least direct links with the other nodes (Figure 7).

The highest mediating role belonged to TP53, and the least mediating role belonged to HDAC1 (Figure 8).

Finally, perhaps the most essential gene in creating the EP300 network and least important was the FOS gene (Figure 9).

### 3.5. Enrichment Analysis

A comprehensive analysis utilizing GO and KEGG was conducted on 309 human genes potentially linked to “AD”, “Inflammation”, “Angiopathy”, and “AFB1”. The GO analysis revealed that these genes are predominantly situated in the extracellular space, nucleus, and cytoplasm concerning cellular components. In terms of molecular functions, they primarily engage in transcription factor binding, DNA-binding transcription factor binding, and cytokine receptor binding. The biological processes associated with these genes include responses to oxygen-containing compounds, regulation of cell death, modulation of molecular functions, positive regulation of gene expression, and signal transduction. Additionally, KEGG enrichment analysis indicated that these proteins are significantly involved in pathways related to cancer, the FoxO signaling pathway, apoptosis, and AD. Furthermore, the diseases correlated with these genes include liver, cardiovascular, and cerebral conditions (Figure 10).

## 4. Discussion

The study explores the complex biochemical pathways and molecular interactions contributing to AD, emphasizing the role of AFB1. AFB1 induces DNA damage, oxidative stress, and endoplasmic reticulum stress. Impaired DNA repair mechanisms, such as those involving BRCA-2 and PARP-1, exacerbate neuronal damage and cognitive decline. AFB1’s impact on key signaling pathways, including those involving PKC, TH, and α-syn, further contributes to neurotoxicity, inflammation, and the formation of Aβ plaques and neurofibrillary tangles. The study underscores the importance of efficient DNA repair, oxidative stress reduction, and targeted therapeutic interventions to prevent or slow AD progression. Bioinformatics analysis reported that key genes such as ESR1, TP53, and EP300 play crucial roles in the pathways. The study highlights the importance of these genes in transcription factor binding, DNA repair, and signal transduction, as well as their involvement in diseases like cancer, cardiovascular conditions, and AD.

AFB1 could affect protein end products and amino acid metabolism, resulting in hyper-ammonemia. This condition easily crosses the blood–brain barrier and triggers the synthesis of glutamate neurotransmitters, which are cytotoxic to brain cells and lead to encephalopathy [52]. One study settles evidence that chronic exposure to AFB1 or dose accumulation leads to neurodegeneration like AD [46] and even cognitive, expressive, and receptive language scores of the study children [53]. Abnormalities in DNA damage repair can be used as diagnostic biomarkers for AD, as well as AFB1-induced cytotoxicity and DNA damage [48]. Substantial DNA damage by AFB1 was associated with the downregulation of *PARP1*, *BRCA2*, and *RAD51* [42], and also with AD [54,55]. Also, AFB1 caused significant DNA damage to neural stem cells [37]. Mutations in DNA repair genes, such as *BRCA2*, have been linked to an increased risk of AD, as they contribute to the accumulation of DNA damage in brain cells [56]. This accumulation can trigger neurodegenerative processes, highlighting the importance of efficient DNA repair in preventing or slowing the progression of AD. Nuclear *PARP1* expression is decreased in AD [57].

AFB1 increased oxidative stress [24,49], as well as led to an increase in ER stress [50]. Genetic and lifestyle-related risk factors for AD are associated with an increase in oxidative stress, suggesting that oxidative stress is involved at an early stage of the pathologic cascade [58,59]. In addition, there was a role of ER stress in the etiology and pathogenesis of AD [60,61].

The activity of the sodium–potassium pump (Na^+^, K^+^-ATPase) was decreased in fish fed a diet contaminated with AFB1 [23,49]. In AD, this pump reduced and caused an excitotoxic cellular response and the resulting neuronal death [62]. Another study [63] showed that the activity of Na^+^, K^+^-ATPase is significantly lower in the brains of patients with AD than in the brains of normal controls.

AFB1 toxicity mechanisms are linked to the production of excessive ROS, upregulated CYP450 activity, oxidative stress, lipid peroxidation, programmed cell death, mitochondrial malfunction, self-digestion, cell death, and inflammatory responses [64]. The cytotoxic action of AFB1 may be due to its ability to interfere with the molecular mechanism of cell cycle regulation such as p53 [47]. Our bioinformatics analysis showed that the role of p53 in cell cycle regulation may be disrupted by the cytotoxic action of AFB1. Levels of p53 are enhanced in the AD brain, maintaining tau hyperphosphorylation, and the interactions of p53 with tau and Aβ represent potential p53-based therapeutics for AD [65]. Also, p53 may be involved in the apoptosis of glial cells in AD brains, but apoptosis in neurons may occur through a p53-independent pathway [66].

AFB1 induced neuroinflammation, trigger the α-synuclein (α-syn) pathology, and caused dopaminergic neurotoxicity [41]. The release of α-syn from damaged presynaptic terminals that occurs during the course of AD was simulated by challenging cells with recombinant α-syn [67]. Recombinant α-syn was shown to directly interact with Aβ1-42 and to decrease the levels of Aβ1-42 oligomers, which might explain its neuroprotective effect [67]. In Parkinson’s disease, α-syn directly regulates dopamine levels [68].

## 5. Conclusions

The study emphasizes that AFB1 contributes to AD by inducing DNA damage, oxidative and ER stress, and impairing DNA repair mechanisms. This leads to neuronal damage, cognitive decline, and neurodegeneration. AFB1 also affects key signaling pathways, decreases sodium–potassium pump activity, and interferes with cell cycle regulation involving p53, contributing to neurotoxicity, inflammation, and the formation of Aβ plaques and neurofibrillary tangles. Efficient DNA repair, oxidative stress reduction, and targeted therapeutic interventions are crucial to prevent or slow AD progression.

### 5.1. Clinical Significance

Understanding the molecular basis of AD, particularly the impact of AFB1, can lead to the identification of potential therapeutic targets. For instance, enhancing DNA repair mechanisms, reducing oxidative stress, and modulating key signaling pathways affected by AFB1 could offer new avenues for treatment. The study also underscores the importance of early detection and intervention to prevent the progression of neurodegenerative processes.

### 5.2. Future Comment

Future research should focus on exploring the therapeutic potential of targeting DNA repair pathways and oxidative stress in AD, with a particular emphasis on mitigating the effects of AFB1. Additionally, investigating the role of specific genes and proteins identified in this study could provide deeper insights into the disease’s pathogenesis. Longitudinal studies and clinical trials will be essential to validate these findings and develop effective treatments for AD.

## Figures and Tables

**Figure 1 biology-14-00436-f001:**
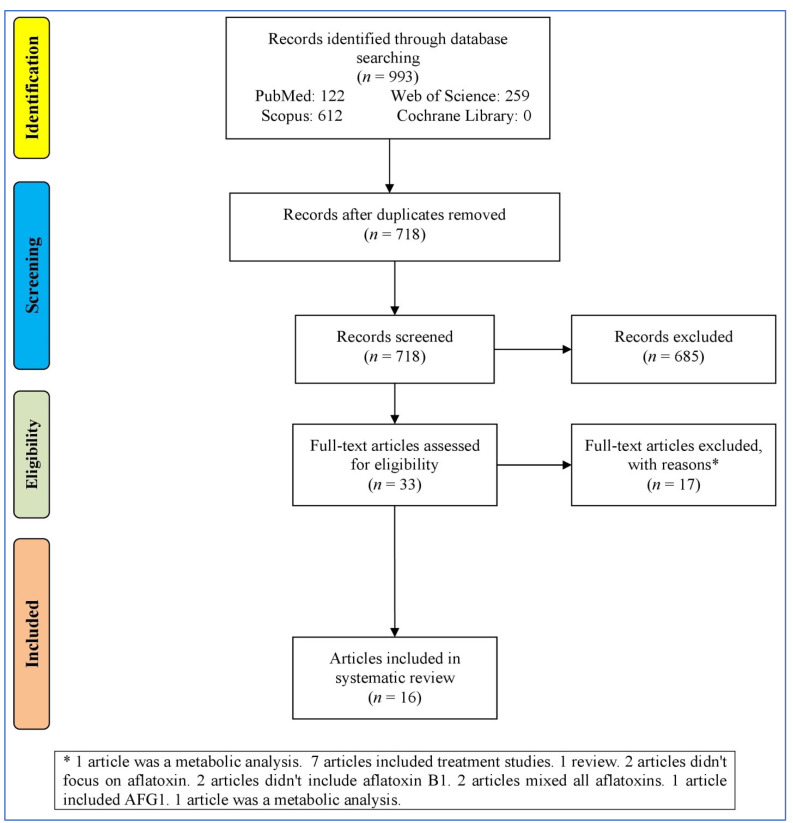
Flowchart of the study selection.

**Figure 2 biology-14-00436-f002:**
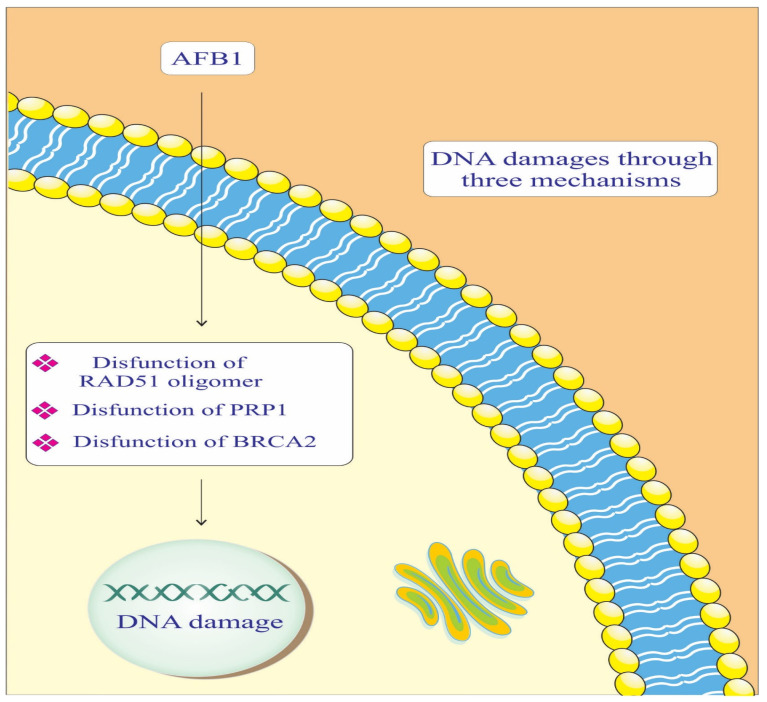
Mechanisms of the effect of AFB1 on DNA repair mediated by RAD-51, BRCA-2, and PARP-1 proteins. PARP-1: Poly [ADP-ribose] polymerase 1. BRCA-2: Breast cancer gene 2. RAD-51: Radiation sensitive protein 51.

**Figure 3 biology-14-00436-f003:**
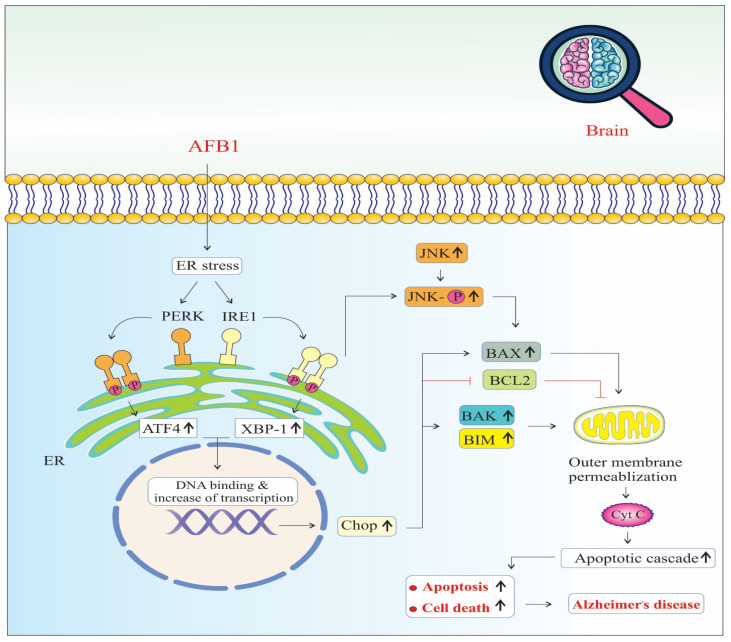
Endoplasmic reticulum stress signaling pathways of AFB1-induced Alzheimer’s disease progression. Chop: The C/EBP homologous protein. BAX: BCL2 Associated X, Apoptosis Regulator. BCL2: B-Cell CLL/Lymphoma 2. BAK: BCL2 Antagonist/Killer 1. BIM: BCL2 Like 11 (Apoptosis Facilitator). JNK: c-Jun N-terminal Kinase (also known as MAPK8). PERK: Protein Kinase R (PKR)-like Endoplasmic Reticulum Kinase (also known as EIF2AK3). IRE1: Inositol-Requiring Enzyme 1 (also known as ERN1). ATF4: Activating Transcription Factor 4. XBP-1: X-Box Binding Protein 1. Cyt C: Cytochrome C, Somatic (also known as CYCS). (↑) means increase.

**Figure 4 biology-14-00436-f004:**
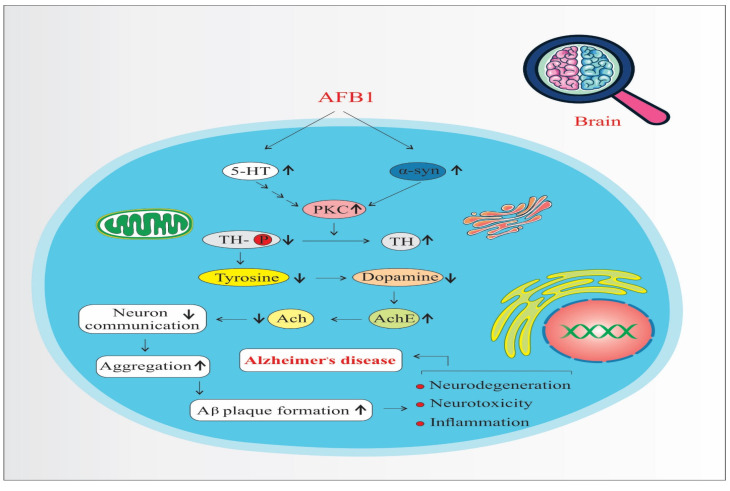
Protein kinase C pathways of AFB1-induced Alzheimer’s disease progression. 5-HT: 5-Hydroxytryptamine (Serotonin). α-syn: Alpha-synuclein. PKC: Protein Kinase C. TH: Tyrosine Hydroxylase. Ach: Acetylcholine. AchE: Acetylcholinesterase. (↑) means increase. (↓) means decrease.

**Figure 5 biology-14-00436-f005:**
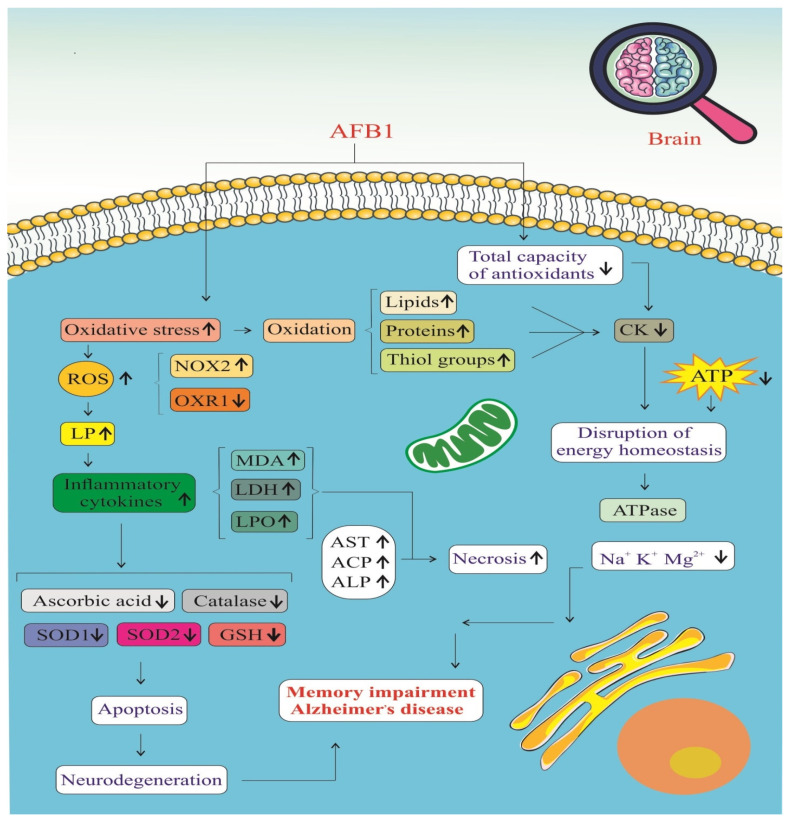
Stress oxidative signaling pathways of AFB1-induced Alzheimer’s disease progression. ROS: Reactive Oxygen Species. NOX2: NADPH Oxidase 2. OXR1: Oxidation Resistance 1. LP: Lipid Peroxidation. MDA: Malondialdehyde. LDH: Lactate Dehydrogenase. LPO: Lipid Peroxidation Products. AST: Aspartate Aminotransferase. ACP: Acid Phosphatase. ALP: Alkaline Phosphatase. SOD1: Superoxide Dismutase 1. SOD2: Superoxide Dismutase 2. GSH: Glutathione. CK: Creatine Kinase. (↑) means increase. (↓) means decrease.

**Figure 6 biology-14-00436-f006:**
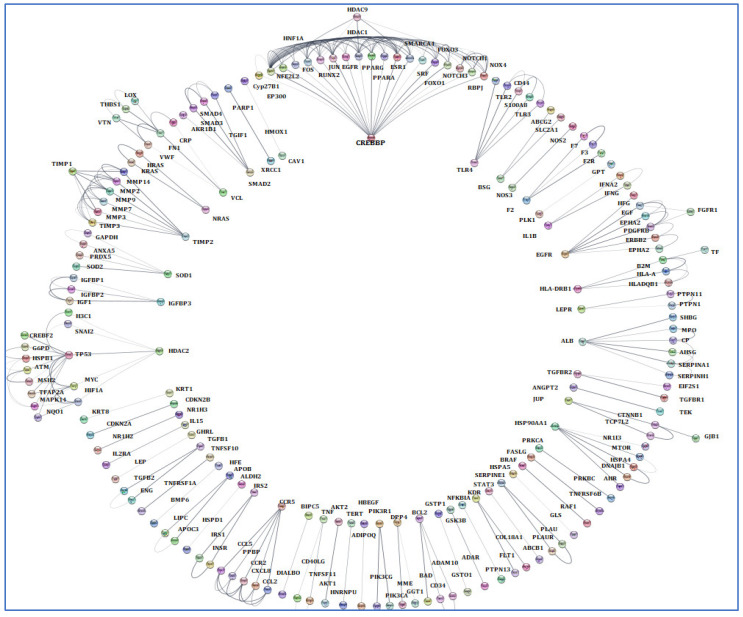
Gene analysis of the identified shared genes was conducted utilizing Cytoscape software 3.6.1. A total of 309 common genes were examined through this platform. The resulting network comprises multiple clusters that exhibit physical interactions among themselves.

**Figure 7 biology-14-00436-f007:**
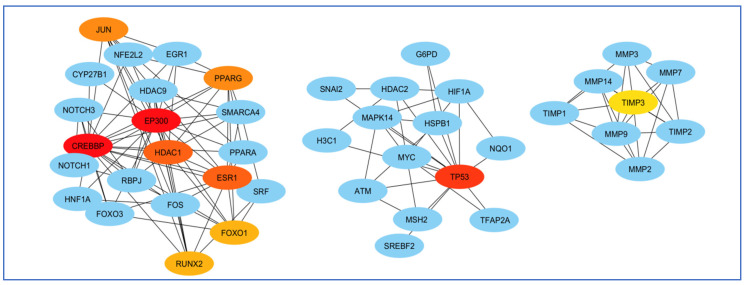
Analysis of the importance of genes based on the total amount of direct links with the other nodes. ESR1 had the most and TIMP3 had the least direct connection with other genes participating in the network. Red colors represent more related genes, and yellow colors represent less related genes.

**Figure 8 biology-14-00436-f008:**
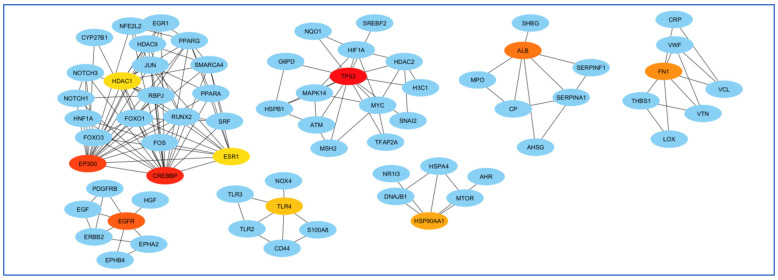
Analysis of genes based on their role in creating communication between genes. The highest mediating role belonged to TP53, and the least mediating role belonged to HDAC1. The red color indicates the more prominent role of that gene in communication.

**Figure 9 biology-14-00436-f009:**
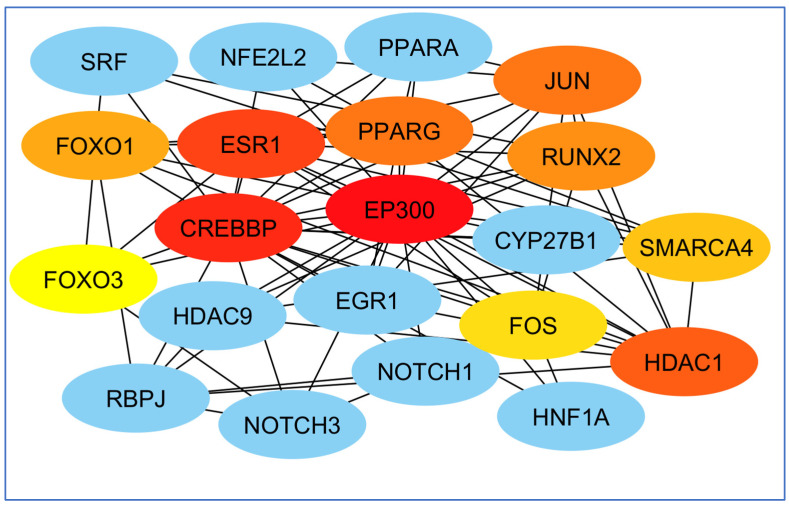
Examination of genes according to their significance in network formation. The EP300 network was primarily influenced by the most critical gene, while the FOS gene was identified as the least significant. The red color denotes a higher level of importance of the gene in the network’s development.

**Figure 10 biology-14-00436-f010:**
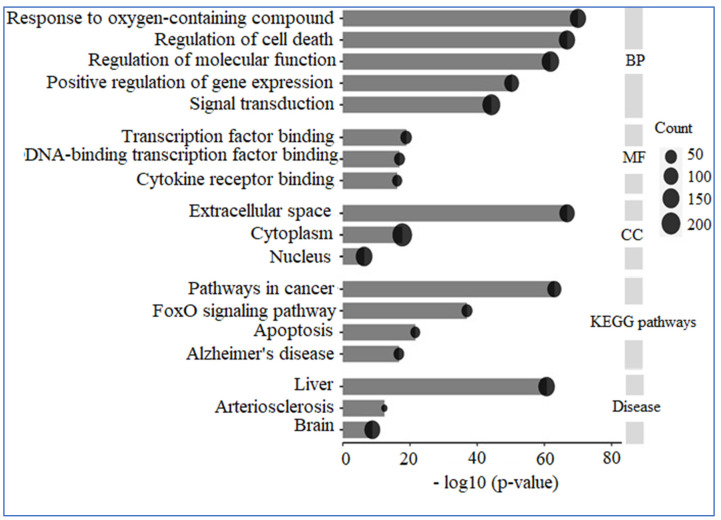
GO and KEGG analysis. The genes under examination primarily participate in biological processes, including the response to reactive oxygen species, the regulation of cell death, and the modulation of molecular functions, among others. Their molecular functions encompass binding to transcription factors, binding to DNA-binding transcription factors, and binding to cytokine receptors. These genes are predominantly located in the extracellular space, cytoplasm, and nucleus. Furthermore, the pathways associated with these genes are linked to conditions such as cancer and AD.

**Table 1 biology-14-00436-t001:** Characteristics of articles included in the systematic review.

The First Author, Publication Year	Country	Type of Cell	Cell Models	Main Results on AFB1-Exposed Brain Cells
Zhou, 2023 [37]	China	Neural stem cell	Human	AFB1 caused significant DNA damage to neural stem cells.
Sahoo, 2024 [39]	India	Glia cells (oligodendrocytes, astrocytes, and ependymal cells)	Goat	The presence of AFB1 toxin in the brain confirms neuroaflatoxicosis in goat kids. AFB1 induces free radicals via the intrinsic pathway of apoptosis. In AFB1-exposed cells: 1. Higher levels of LPO. 2. Reduced levels of antioxidant enzymes (Catalase, SOD, GSH). 3. Strong immunoreactivity of 8-OHdG in the brain, indicating high oxidative stress. 4. Increased immunosignaling of caspase-3 and caspase-9 in the brain, suggesting an association with the intrinsic pathway of apoptosis.
Almanaa, 2024 [38]	Saudi Arabia	NR	Mouse	AFB1 increases mRNA expression levels of: Notch-1, IL-6, MCP-1, iNOS, GM-CSF, NF-κB p65 in the brain. Exposure to AFB1 exacerbates immunological abnormalities by increasing the expression of inflammatory mediators.
Alwetaid, 2024 [40]	Saudi Arabia	NR	Mouse	AFB1 increased mRNA expression levels of CCR3, CCR7, CCR9, CXCR3, CXCR4, and CXCR6.
Wang, 2023 [41]	USA	Microglial	Mouse	AFB1 induced neuroinflammation, triggered the α-synuclein pathology, and caused dopaminergic neurotoxicity. AFB1 increases soluble epoxide hydrolase in the brain.
Baldissera, 2018 [23]	Brazil	Neuron	Fish	AChE activity in brain synaptosomes was increased in fish fed a diet contaminated with AFB1. Activity of the sodium–potassium pump (Na^+^, K^+^-ATPase) was decreased in fish fed a diet contaminated with AFB1. This diet (1177 ppb kg feed^−1^) caused a disruption of the BBB and brain lesions, which may contribute to the behavioral changes.
Alwetaid, 2023 [43]	Saudi Arabi	NR	Mouse	AFB1 resulted in a significant upregulation of various immune-related factors, including *IFN-γ*, *STAT1*, *T-bet*, *IL-9*, *IRF4*, *IL-17A*, *IL-21*, *RORγ*, *STAT3*, *IL-22*, *AhR*, and *TNF-α*. The exposure to AFB1 resulted in a significant downregulation of *IL-10, TGF-β1*, and *FoxP3* by CD4+ T cells.
Huang, 2020 [42]	China	Neuroblastoma	Mouse	ROS levels in IMR-32 cells increased significantly in a time- and AFB1 concentration-dependent manner, which was associated with the upregulation of *NOX2*, and downregulation of *OXR1*, *SOD1*, and *SOD2*. Substantial DNA damage associated with the downregulation of *PARP1*, *BRCA2*, and *RAD51* was also observed. AFB1 significantly induced S-phase arrest, which is associated with the upregulation of *CDKN1A*, *CDKN2C*, and *CDKN2D*. AFB_1_ induced apoptosis involving *CASP3* and *BAX*.
Bonsi, 1999 [44]	Italy	Neuroblastoma	Mouse	AFB1 is an inhibitor of cyclic nucleotide phosphodiesterase activity. AFB1 significantly inhibits cAMP and cGMP hydrolytic activity.
Vahidi-Ferdowsi, 2018 [45]	Iran	Astrocyte	Mouse	AFB1 (32 nM) induces apoptosis in astrocytes through ATP depletion and caspases activation.
Alsayyah, 2019 [46]	Saudi Arabia	Astrocyte	Rat	AFB1 results in several pathophysiological circumstances in a duration-dependent manner concerning neurodegeneration, especially Alzheimer’s disease. Acid phosphatase, alkaline phosphatase, aspartate aminotransferase, lactate dehydrogenase activities, and lipid peroxidation increased by AFB1. AFB1 reduced brain Activating Transcription Factor (CK), SOD1, SOD2, catalase, glutathione, and glutathione peroxidase, leading to necrosis.
Ahmed, 1984 [51]	India	NR	Chicken	AFB1 increased 5-HT and reduced norepinephrine.
Ricordy, 2002 [47]	Italy	Neuroblastoma (SK-N-SH)	Human	The cytotoxic action of AFB1 may be due to its ability to interfere with the molecular mechanism of cell cycle regulation, such as p53 and p21.
Zheng, 2018 [48]	China	Neuroblastoma (SK-N-SH)	Human	AFB_1_ induced cytotoxicity and DNA damage. AFB_1_ significantly inhibited cell growth, elevated the level of lactate dehydrogenase, induced genetic damage, and increased the levels of ERK1/2 and JNK.
Souza, 2019 [49]	Brazil	NR	Silver catfish	AFB1 inhibits cerebral CK activity, as well as the involvement of oxidative stress on this inhibition. AFB1 reduced the brain sodium–potassium pump (Na^+^, K^+^-ATPase) activity.
Song, 2024 [50]	China	Neuron	Mouse	AFB1 led to an increase in ER stress, as indicated by an increase in the phosphorylation of *IRE1*, *PERK*, and *EIF2a*, as well as an increase in the expression of the *Chop*, *Xbp1s*, and *Atf4* genes, leading to apoptosis.

NR: Not reported. AFB1: Aflatoxin B1, LPO: Lipid Peroxidation, SOD: Superoxide Dismutase, GSH: Glutathione, 8-OHdG: 8-Hydroxy-2’-deoxyguanosine, IL-6: Interleukin 6, MCP-1: Monocyte Chemoattractant Protein-1, iNOS: Inducible Nitric Oxide Synthase, GM-CSF: Granulocyte-Macrophage Colony-Stimulating Factor, NF-κB p65: Nuclear Factor kappa-light-chain-enhancer of activated B cells p65 subunit, CCR3: C-C Chemokine Receptor Type 3, CCR7: C-C Chemokine Receptor Type 7, CCR9: C-C Chemokine Receptor Type 9, CXCR3: C-X-C Chemokine Receptor Type 3, CXCR4: C-X-C Chemokine Receptor Type 4, CXCR6: C-X-C Chemokine Receptor Type 6, sEH: Soluble Epoxide Hydrolase, AChE: Acetylcholinesterase, Na+, K+-ATPase: Sodium–Potassium Adenosine Triphosphatase, BBB: Blood–Brain Barrier, IFN-γ: Interferon Gamma, STAT1: Signal Transducer and Activator of Transcription 1, T-bet: T-box Transcription Factor TBX21, IL-9: Interleukin 9, IRF4: Interferon Regulatory Factor 4, IL-17A: Interleukin 17A, IL-21: Interleukin 21, RORγ: RAR-related Orphan Receptor Gamma, STAT3: Signal Transducer and Activator of Transcription 3, IL-22: Interleukin 22, AhR: Aryl Hydrocarbon Receptor, TNF-α: Tumor Necrosis Factor Alpha, IL-10: Interleukin 10, TGF-β1: Transforming Growth Factor Beta 1, FoxP3: Forkhead Box P3, ROS: Reactive Oxygen Species, NOX2: NADPH Oxidase 2, OXR1: Oxidation Resistance 1, SOD1: Superoxide Dismutase 1, SOD2: Superoxide Dismutase 2, PARP1: Poly (ADP-Ribose) Polymerase 1, BRCA2: Breast Cancer Type 2 Susceptibility Protein, RAD51: DNA Repair Protein RAD51, CDKN1A: Cyclin Dependent Kinase Inhibitor 1A, CDKN2C: Cyclin Dependent Kinase Inhibitor 2C, CDKN2D: Cyclin Dependent Kinase Inhibitor 2D, CASP3: Caspase-3, BAX: Bcl-2-Associated X Protein, PDE: Phosphodiesterase, cAMP: Cyclic Adenosine Monophosphate, cGMP: Cyclic Guanosine Monophosphate, ATP: Adenosine Triphosphate, ACP: Acid Phosphatase, ALP: Alkaline Phosphatase, AST: Aspartate Aminotransferase, LDH: Lactate Dehydrogenase, CK: Creatine Kinase, GPx: Glutathione Peroxidase, 5-HT: 5-Hydroxytryptamine, ERK1/2: Extracellular Signal-Regulated Kinases 1 and 2, JNK: c-Jun N-terminal Kinase, ER: Endoplasmic Reticulum, IRE1: Inositol-Requiring Enzyme 1, PERK: Protein Kinase RNA-like Endoplasmic Reticulum Kinase, EIF2a: Eukaryotic Translation Initiation Factor 2 Alpha, Chop: C/EBP Homologous Protein, Xbp1s: Spliced X-box Binding Protein 1, Atf4: Activating Transcription Factor 4, p53: Tumor Protein p53, and p21: Cyclin Dependent Kinase Inhibitor 1.

## Data Availability

Data sharing is not applicable to this article as no datasets were generated or analysed during the current study.

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
