# Peer review of "Effect of Aflatoxin B1 on the Nervous System: A Systematic Review and Network Analysis Highlighting Alzheimer’s Disease"

_biology, 2025, doi:10.3390/biology14040436_

Round 1
Reviewer 1 Report
Comments and Suggestions for Authors
All the comments are included in the attacched document "Comments to the authors_Biology"

Written in clear academic English, but some modifications could improve understanding.
Author Response
We thank Reviewer #1 for the valuable comments, which helped us to improve the quality of the revision.
Please find attached the detailed point-by-point-response.
Thank you once again for all your kind efforts.

Reviewer 2 Report
Comments and Suggestions for Authors
This manuscript reviewed literature to examine the relationship between aflatoxin exposure and Alzheimer's Disease.
The review's language was appropriate and well written. There were several minor points of comment for correction.
Minor points
Line 82 – This idea should be clarified. The behavioral effects are separate from the BBB penetration which is not itself a behavioral effect. Also it would be best to note this data was from a fish model and not humans as the comments directly before it relate to human outcomes.
Line 84 Again it would be best to state the data from cell based experiments supports the hypothesis that these are the mechanisms of action.
Please clearly state what type of model or assay the evidence comes from in each case to support the hypothesized mechanisms.
Line 87 The use of ‘unnecessary ‘ should be removed. There is a determination if the inflammation is necessary or not only that one of the consequences of it is the production of free radicals etc.
Line 88/89 ‘This leads to chronic inflammation, cancer, and neurodegenerative diseases (22, 23). ‘
This comment is very overstated given the citations used to support it. It may play a role in the development of cancer and neurodegenerative disease but more specific citations should be used to claim this.
Line 90/91 “AF-protein adducts are most frequently associated with acute intoxication, as they block protein synthesis, particularly the enzymes involved in vital functions such as metabolic pathways, protein synthesis, DNA replication and repair, and immune response”
This manuscript would be greatly improved if there was original research cited to support each of the items claimed in the above sentence.
Line 105 “Can AFB1 effect on nervous system especially AD?”
Th grammar of this sentence should be corrected. It is concerning that this was submitted as the PECO question as the basis of the study.
Line 135 It may be best to add the exclusion reasons to the text in addition to the legend of the figure. Also mention the exclusion criteria for the screening of the 718 papers.
Figure 2.
The figure overlay implies that the mechanisms occur in specific brain regions but are actually general. The depiction of these should be altered to avoid this confusion.
Line 157 /158 “Terms like “ATP depletion", “cytotoxicity", “lipid peroxidation", "neurotoxicity", "neuroinflammation", "apoptosis", "apoptosis", “oxidative stress”, etc. are interconnected.
This statement is vague and does not add to the manuscript -should likely be removed.
Line 314 why is neuronal death underlined?
Major issues
The most problematic issue was the lack of citing original research for the 3.3 Pathways section. There was not a single citation.
The figures 2 and 3 were broad and indistinct in what they impart. They should be substantially revised or removed.
Author Response
We thank Reviewer #2 for the valuable comments, which helped us to improve the quality of the revision.
Please find attached the detailed point-by-point-response.
Thank you once again for all your kind efforts.

Reviewer 3 Report
Comments and Suggestions for Authors
I am offering my deep gratitude to you for inviting me to review the manuscript entitled “Effect of Aflatoxin B1 on the Nervous System: A Systematic Review and Network Analysis Highlighting Alzheimer's Disease.” Generally, the manuscript provides an interesting review of one of the most toxic mycotoxins. Upon carefully reviewing the manuscript, I found that all parts of it were written in a scientifically sound manner, and the experiments were conducted professionally. The results have been summarized and discussed satisfactorily. However, there are some minor remarks that need to be addressed before the manuscript can be accepted for publication in the Biology journal.
I hope these suggestions will help improve the scientific value of the manuscript:
- Key words should be listed in alphabetical order.
- The abbreviations used in the manuscript should be revised.
- The resolution of Figure 11 should be checked.
Decision: The manuscript can be published in the Journal of Biology-MPDI Journal after addressing the remarks mentioned above as a minor revision.
Comments on the Quality of English LanguageThe language is good
Author Response
We thank Reviewer #3 for the valuable comments, which helped us to improve the quality of the revision.
Please find attached the detailed point-by-point-response.
Thank you once again for all your kind efforts.

Round 2
Reviewer 1 Report
Comments and Suggestions for Authors
All the previous comments have been taken into account.
L335: revise an extra space
Author Response
Dear Reviewer,
Thank you once again for your scrutiny devoted to the present manuscript. Please find attached the detailed point-by-point-response.
Thank you once again for your work!
Sincerely

Reviewer 2 Report
Comments and Suggestions for Authors
- The statement linking aflatoxin to free radical production has no citation.
References 25-29 reference only free radicals causing the pathologies indicated. None of these are specific to aflatoxin and without the previous citation are lacking relevance.
- This reviewer finds it difficult to have the PECO question edited for the manuscript and read that there were no results based on the question (Line 108). The editing of the question is disingenuous if it was not resubmitted, and the Prospero Database shows the original question
https://www.crd.york.ac.uk/PROSPERO/view/CRD420250651007
accessed 4.6.2025
“Review objectives
Can AFB1 effect on nervous system especially AD?”
At a minimum that authors should submit their edited question to the Prospero Database and report if it found additional studies to include and convey this to the editor and address this idea in the discussion of the manuscript.
The poor grammar of the original question could have limited the scope of the results. The results of this search are the fundamental basis of this review.
This reviewer still finds this a substantial issue in the appropriateness of this manuscript.
- “Recent studies have investigated the effects of AFB1 on brain cells across different countries (study locations), revealing significant neurotoxic effects on various cell types (such as neurons, astrocytes, and microglia) and specific cell models (human, mouse, fish, and livestock models) as both in vivo and in vitro (Table 1)”
Also, the edit above does not meet the meaning of the review point. It is the point that the ‘neurotoxic effects on various cell types (such as neurons, astrocytes, and microglia)’ were discovered in non-primate animal models. This is significantly important when followed by a statement that human cell lines were also used and combining these ideas as in vivo and in vitro.
Author Response

(The authors gave the same response as above.)
